# Controlling gene expression timing through gene regulatory architecture

**Md Zulfikar Ali** [1,2], **Robert C. Brewster** [1,2] *

**1** Department of Systems Biology, University of Massachusetts Chan Medical School, Worcester, Massachusetts, United States of America, **2** Department of Microbiology and Physiological Systems, University of Massachusetts Chan Medical School, Worcester, Massachusetts, United States of America

* Robert.Brewster@umassmed.edu

**Data Availability Statement:** The source code for generating data from this work can be downloaded from https://github.com/zulfikgp/Timing.

**Funding:** Research reported in this publication was supported by NIGMS of the National Institutes of Health (https://www.nigms.nih.gov/) under award

## Abstract

Gene networks typically involve the regulatory control of multiple genes with related function. This connectivity enables correlated control of the levels and timing of gene expression. Here we study how gene expression timing in the single-input module motif can be encoded in the regulatory DNA of a gene. Using stochastic simulations, we examine the role of binding affinity, TF regulatory function and network size in controlling the mean first-passage time to reach a fixed fraction of steady-state expression for both an auto-regulated TF gene and a target gene. We also examine how the variability in first-passage time depends on these factors. We find that both network size and binding affinity can dramatically speed up or slow down the response time of network genes, in some cases predicting more than a 100-fold change compared to that for a constitutive gene. Furthermore, these factors can also significantly impact the fidelity of this response. Importantly, these effects do not occur at "extremes" of network size or binding affinity, but rather in an intermediate window of either quantity.

## Author summary

Regulated genes are able to respond to stimuli in order to ramp up or down production of specific proteins. Although there is considerable focus on the magnitude (or fold-change) of the response and how that depends on the architectural details of the regulatory DNA, the dynamics, which dictates the response time of the gene, is another key feature of a gene that is encoded within the DNA. Unraveling the rules that dictate both the response time of a gene and the precision of that response encoded in the DNA poses a fundamental problem. In this manuscript, we systematically investigate how the response time of genes in auto-regulatory networks is controlled by the molecular details of the network. In particular, we find that network size and TF-binding affinity are key parameters that can slow, in the case of auto-activation, or speed up, in the case of auto-repression, the response time of not only the auto-regulated gene but also the genes that are controlled by the auto-regulated TF. In addition, we find that the precision of the response depends crucially on these characteristics.

R35GM128797 (RCB). The funders had no role in study design, data collection and analysis, decision to publish, or preparation of the manuscript.

**Competing interests:** The authors have declared that no competing interests exist.

## Introduction

Transcription factors (TFs) play an essential role in controlling how a cell will respond to a stimulus through the regulation of gene expression [1–3]. The gene regulatory code, inscribed in the DNA of each gene, specifies how the production of each gene product will be controlled in space, time and magnitude. A tremendous effort has focused on which regulatory proteins (TFs) regulate each gene, where they bind to accomplish their regulation and the mechanism by which the TF functions [4–9]. While this work has advanced our ability to read the regulatory code, an important consideration is how the timing of gene expression may also be encoded in the DNA. Although temporal patterns in expression can be arranged through a variety of mechanisms such as signalling [10, 11] or compartmentalization [12–14], here we will focus on timing encoded through transcriptional regulation. TFs are often part of a "single-input module" network motif where one auto-regulatory TF gene controls many genes [15, 16]; typically these target genes are consecutive enzymes along a single physiological pathway and their temporal order in expression is crucial to ensure efficient response [17–20]. However, despite the prevalence of this regulatory motif in natural genetic circuits and the importance of timing for efficient transcriptional programs, predicting the timing of gene expression from this network motif based on regulatory sequence is challenging.

The natural timescales to reach steady-state of gene regulatory systems is set by the decay rate of the protein product [21]. In bacteria such as *E. coli*, where most proteins are stable, the timescale is set by the division rate of the cell while higher organisms tend to have shorter lived proteins. Regardless of organism, natural regulatory systems operate at a wide range of timescales sometimes requiring precise tuning [17, 22]. The timing, and variability in timing, of cellular events such as metabolism and DNA damage response in *E. coli* [17, 18], meiosis in *S cerevisiae* [23], sporulation in *B. subtilis* [24] and apoptosis in human cell lines [25] have been identified as phenomena whose temporal dynamics are driven by transcriptional regulation. Recent studies have highlighted the importance of network motifs in temporal filtering and the coordination of events [26] as well as the role of interaction affinities and TF cooperativity in tuning gene circuit timing [27]. The network architecture will play a role in setting these timescales and the accompanied noise [28]. It is known that auto-regulation can slow down (in the case of activation) or speed up (repression) the response time of an autoregulated gene [21, 29, 30]. Furthermore, target genes in the network are impacted by these dynamics, each target gene in a network may have different regulatory DNA which may include differing number of TF binding sites, affinity for the TF or interactions with other TFs at those sites. Some studies have pointed to differential TF binding affinity at each target gene within a regulatory network as a driving force in setting timescales of expression [17, 18, 31–36]. However, other factors such as the size of the network (number and affinity of additional binding sites) likely play a significant role; TFs can target dozens to hundreds of specific sites in the genome and the presence of competitive sites can have a significant impact on the steady-state level of expression and noise of the genes in the network [37–43]. Exploring the role regulatory architecture and competing binding sites play in tuning not only the average response time but also the precision of that response is an open question.

Here, we use stochastic simulations to characterize how a multitude of factors control the timing of gene expression [44]. The primary focus of this work is auto-regulated TF genes and the target genes they regulate. This network motif is common for TF genes, for instance the majority of TFs in *E. coli* are auto-regulated, and as such it is crucial to study this motif and its role in regulating the dynamics of gene expression of both itself and the other genes under its control within the network [4, 5]. We start by examining auto-regulated genes alone and explore how details of the regulation such as binding affinity, TF regulatory role (activation or

repression) and network size (number of TF binding sites) influence the mean first-passage time (MFPT) to reach 50% of the steady state expression level. Importantly, the MFPT is a single-cell measurement of response time (*i.e.* it is the average over many single cell trajectories); as opposed to measuring how long the bulk takes to reach a certain expression level on average; the average time it takes cells to reach a threshold and the time it takes a bulk to reach a threshold are different quantities. MFPT has been widely used to study biological processes [29, 45–48]. Furthermore, we also examine the robustness of the response; *i.e.* the variability in first-passage time at the single cell level; auto-regulation is known to impact the stability and noise of expression [49, 50], here we explore how these elements influence the variability of response of a gene within the network. Although we approach this problem in the context of bacteria, in order to understand how a cell may produce different temporal responses when most proteins share the same degradation rate, we also demonstrate that our results are general and qualitatively robust to parameter ranges more typical of Eukaryotic systems as well.

## Results

### Model

The general model of gene expression for a self regulatory gene (TF-gene) is outlined in Fig 1A. In this model, the TF-gene produces TFs that can bind to the promoter of the TF-gene with probability set by the binding-rate ($k_{on}$) and assumed to be diffusion limited and is thus kept constant [51, 52]. The TF binding affinity is controlled through the unbinding rate ($k_{off}$), which is related to the sequence specificity of the binding site; higher affinity means lower unbinding rate and vice versa [53]. The regulatory function of the TF is represented as a change to transcription rate when the TF is bound; the TF-gene transcribes at rate $r$ when bound by a TF and $r_0$, when unoccupied. The ratio of the transcribing rates ($\alpha = r/r_0$) for the TF-gene determines whether the gene is self-activating ($\alpha > 1$), self-repressing ($\alpha < 1$) or constitutive ($\alpha = 1$). In later sections, we add the possibility of decoy TF binding sites to the model. Outlined in Fig 1B, decoy TF binding sites bind a TF (thus making it unavailable to bind to the TF gene) but do not result in regulation (or do not regulate a gene whose product we consider). Decoy sites have the same TF binding rate as the TF gene but can have a distinct unbinding rate, $k_{off,d}$. For the sake of simplicity we consider $N$ identical decoy sites, though in a real network these sites could have a spectrum of affinities. Finally, as outlined in Fig 1C, we will explore the dynamics of a target gene. As is the case for decoys, the TF binding site for this gene binds TFs with the same rate as every other binding site and has a distinct unbinding rate, $k_{off,t}$. We also consider that the regulatory role of the TF may be different on this target gene and thus we define $\alpha_t$ as the ratio of TF-bound ($r_t$) over TF-unbound transcription rate ($r_{0,t}$) of this gene (see Materials and methods for details and parameter values). Notably, TF target function and auto-regulatory function can be different. Fig 1D shows the number of activating (blue bars) and repressing (red bar) targets for 74 autoregulated TFs in *E. coli*. Clearly, this demonstrates that a TFs function at its target genes can be different from its auto-regulatory function; *i.e.* a TF may repress its own expression while activating its targets or vice versa. As such, we allow the regulatory strength of the TF on the target gene to be different form the regulatory strength of that same TF on itself.

In our model, we assume protein degradation as random events with half-life equal to the cell-division time. However, inclusion of an explicit cell-division process (where each protein degrades with 50% probability every cell division event) does not alter our findings, see S1 Text and Fig 1 in S1 Text.

To explore the dynamics of this system, we employ a stochastic simulation algorithm to compute mean first passage time (MFPT) for gene expression to reach half of the steady state

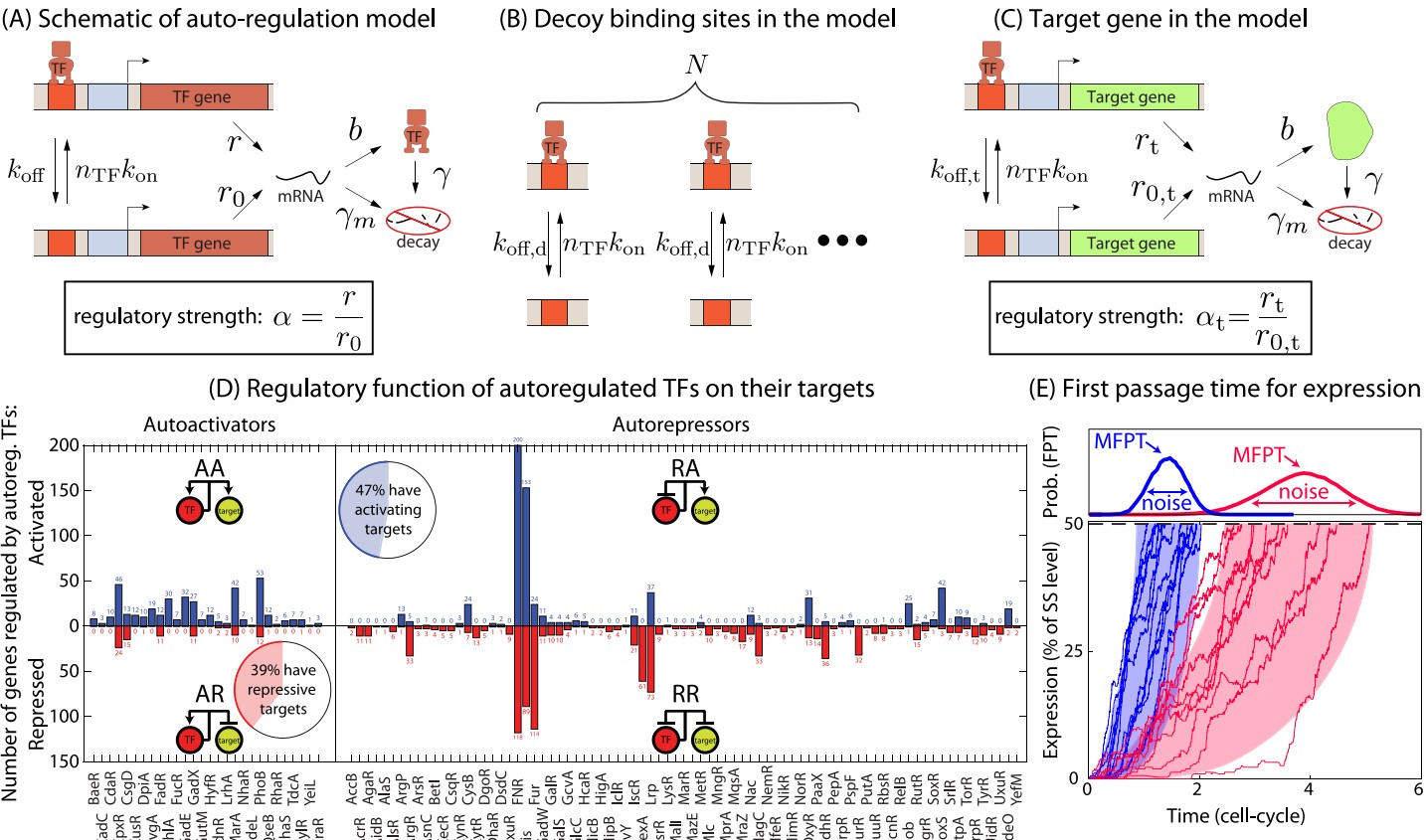

**Fig 1. Mean first passage time in a self regulated gene.** (A) Schematic of a self-regulating gene. (B) Schematic showing the reactions of TF binding/unbinding to competing decoy sites. (C) Reactions of a target gene regulated by the auto-regulatory TF gene. (D) Number of genes activated (blue bars) or repressed (red bars) by some known autoregulated TFs in *E. coli*. Data is taken from RegulonDB [54]. (E) First passage time is defined as the time for the gene expression to hit the threshold for the first time as shown here. For an auto-regulated gene, the mean first passage time (MFPT) and the distribution of FPT strongly depend on the strength of auto-regulation ($\alpha$) and the TF-binding affinity.

expression level [45, 55]. The choice of threshold, 50% of the steady state expression, is arbitrary, but consistent with other work in the field [30, 56, 57], and the consequence of this choice is also discussed at points below. As demonstrated schematically in Fig 1E, at time $t = 0$ the network is switched on and the genes start to express until the threshold is achieved. The time to reach the threshold is termed as "first passage time" (FPT) and is computed for each of $> 10^5$ individual trajectories (cells). The MFPT is obtained by taking the mean of the FPT distribution. We also calculate the noise in the timing of response represented by the coefficient of variation (CV) which is the ratio of the standard deviation to the mean of the FPT distribution.

## Role of TF-binding affinities in controlling expression timing of auto-regulated genes

We start by investigating the impact of TF-binding affinity in auto-regulatory motifs without any competing binding sites (decoy, Fig 1B) or target genes (Fig 1C). For both positive and negative auto-regulation motifs, the TF-binding affinity ($k_{off}$) impacts the mean first passage time (MFPT) to reach half of the Steady-state expression level (SSE) in a non-monotonic way. In Fig 2A, we show the dependence of the MFPT on the binding affinity of a positive auto-

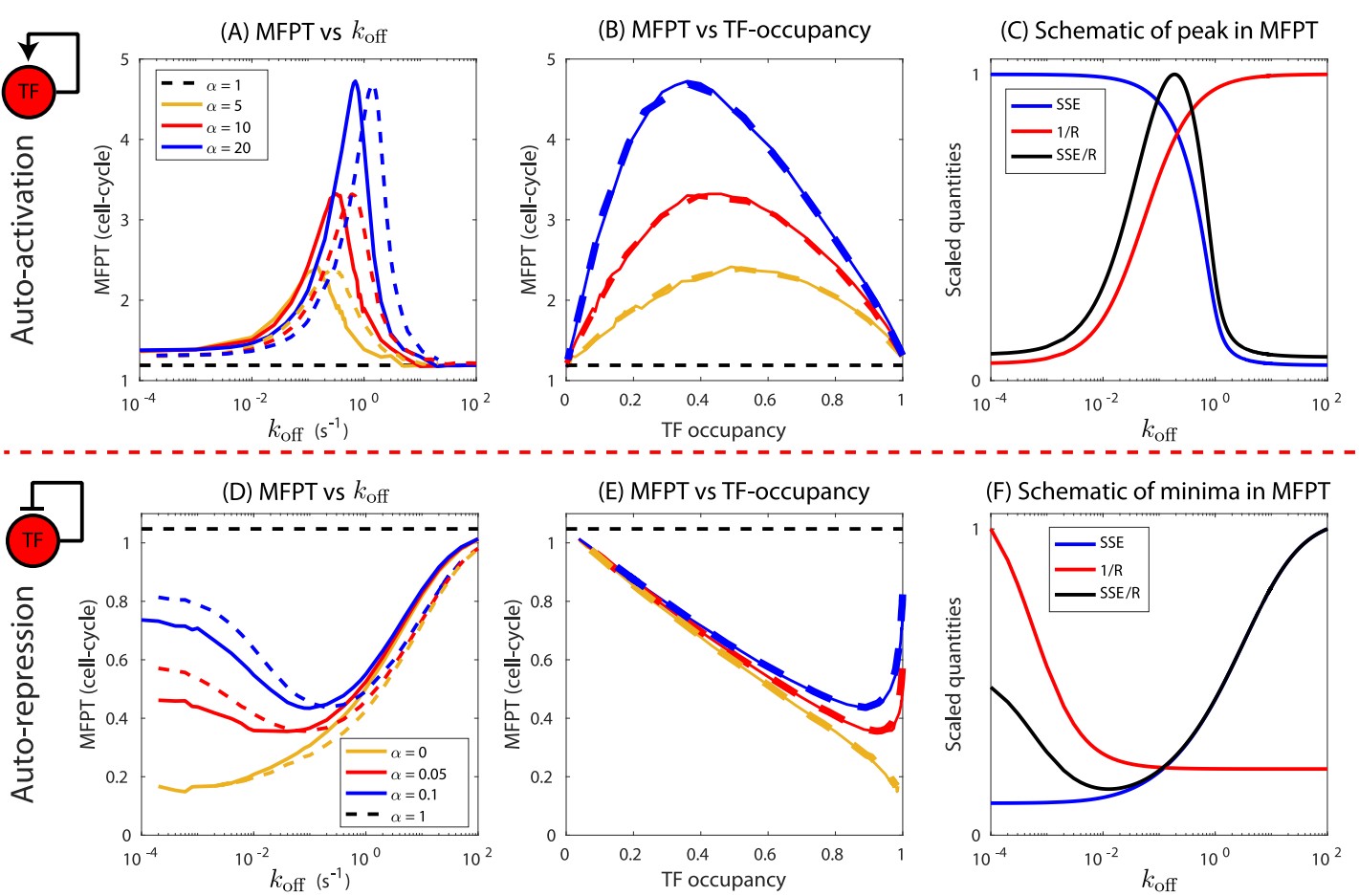

**Fig 2. Effect of TF-binding affinity on MFPT for auto-regulated genes without any competing binding sites.** (A, D) MFPT as a function of off-rate (binding affinity) of auto-activated gene and auto-repressed gene. Each curve (blue, red, yellow) is generated by keeping the transcription rate ($r_0$) and translation rate ($b$) fixed and varying only the binding affinity through $k_{off}$. Dashed curves correspond to doubling the translation rate ($2b$). Black dashed lines correspond to MFPT of a constitutive gene of transcription rate $r_0$. (B, E) MFPT as a function of TF-promoter occupancy. Solid and dashed curves are for translation rates $b$ and $2b$, respectively. (C,F) Scaled steady state TF expression (SSE) and inverse of production rate ($1/R$) as a function of off-rate. The ratio of SSE and $R$ (black curve) qualitatively predicts the behavior of the MFPT as a function of off-rate. For auto-activation we use $r_0 = 0.0025$ s$^{-1}$, $b = 0.025$ s$^{-1}$mRNA$^{-1}$; for auto-repression $r_0 = 0.05$ s$^{-1}$, $b = 0.1$ s$^{-1}$mRNA$^{-1}$. The protein and mRNA degradation rates are $\gamma = 0.0003$ s$^{-1}$ and $\gamma_m = 0.011$ s$^{-1}$ per molecule, all additional parameter values listed in the Materials and methods.

regulatory gene. In the two extreme limits of affinity, the MFPT is a minimum approximately equal to that for a constitutive gene (set by the transcription and translation rates, roughly equal to one cell-cycle here; see S1A Fig). When the binding affinity is very weak, the promoter is always free and expresses with a lower basal transcription rate $r_0$ and thus the MFPT is essentially the same as a constitutive gene with transcription rate $r_0$. On the other hand, when the binding affinity is strong, the promoter is always occupied by a TF and as such the MFPT is similar to a constitutive gene with transcription rate $r$. However, the MFPT is slightly higher than that of a constitutive gene due to the fact that the first binding event (to initiate the auto-activation) takes additional time and thus the MFPT is the sum of the MFPT of a constitutive gene and the average time to express the first TF to initiate the auto-activation.

Interestingly, between these two extremes, the MFPT has an intermediate maximum at a specific value of TF binding affinity ($k_{off}$). The height of this intermediate maximum depends on the strength of the activation ($\alpha = r/r_0$), with lower strength corresponding to smaller maxima. This maxima indicates that in a specific range of binding affinity, the auto-activating TF

gene can respond substantially slower than a constitutive gene, taking several generations to respond and reach the half maxima of steady-state expression. For a given $\alpha$, altering translation rate shifts the position of maxima without affecting the peak MFPT (Fig 2A, dashed curves). This shift vanishes if we replot the data as a function of steady state TF-occupancy, the fraction of time promoter is occupied (see Materials and methods), rather than $k_{off}$ (Fig 2B); when plotted as a function of TF-occupancy the data for different translation rates collapses such that the maxima occurs at a specific value of TF occupancy for a given $\alpha$ regardless of the translational dynamics. However, the occupancy level where the maxima occurs still has a dependence on the strength of regulation, $\alpha$.

We suspected the non-monotonic behavior of the MFPT for the auto-activating gene is the result of interplay between the steady state expression level (SSE) and the effective TF-production rate. While the SSE level can be precisely determined from the simulation, the effective TF-production rate, on the other hand, has a complex dependence on time and changes as the number of TFs increases. In order to gain an intuition for the non-monotonic behavior of the MFPT, we use a simplified TF-production rate: the TF-production rate from a single TF ($R$). In this case, SSE is proportional to how much TF must be made by the gene to reach the FPT threshold and $R$ is an approximation for the speed in which the system approaches that level (SSE has units of number of proteins and $R$ is a rate of production given by $R = br_0 + b(r - r_0) k_{on}/(k_{on} + k_{off})$). As such, the ratio of these quantities (SSE/$R$) should tell us approximately the timescale of response of the gene. For the auto-activating gene, Fig 2C shows the steady state expression level (blue curve) and the quantity $1/R$ (red curve) as a function of TF binding affinity, $k_{off}$. The SSE level decreases as $k_{off}$ is increased, from a maximal value of SSE = $rb/\gamma_m \gamma$ for very strong activation when the promoter is always occupied by a TF to SSE = $r_0b/\gamma_m \gamma$ for very weak activation. The production rate $R$ behaves in a similar fashion and we plot $1/R$ as a red curve on the same axis. The response time, which is then the product of the red and blue curves (SSE/$R$) will be non-monotonic with an intermediate maximum that approximately coincides with the maximum of the MFPT; although the maximum does not exactly coincide due to the simplifications in this calculation. Furthermore, a deterministic toy-model (see S3 Text) finds an analytical expression for response time ($\tau$) analogous to MFPT which is given by

$$\tau = -\frac{\ln(1-f)}{\gamma} + \frac{B}{\gamma\sigma}\left[\ln(1-f) - \ln\left(1 + f\frac{a+b}{b-a}\right)\right].$$

Here, $f$ is the expression threshold and $\sigma = k_{on}/k_{off}$. $a$, $b$ and $B$ are functions of the model parameters (see S3 Text). The first term on the right hand side, $-\ln(1-f)/\gamma$, is the response time for a constitutive gene and depends only on the threshold and protein degradation rate. The analytical solution shows that the response time $\tau$ of an auto-regulated gene can be higher or lower than that of a constitutive gene. Furthermore, it yields the same non-monotonic behavior of response time as a function of $k_{off}$ (see Fig 1A in S3 Text).

As shown in Fig 2D, we see a similar phenomenon for an auto-repressing gene. In all cases, the MFPT is a minimum value, less than a cell-cycle [30], when the TF off-rate is very low (high affinity) and approaches the MFPT of a constitutive gene when the site is very weak. For auto-repression we once again see non-monotonic behavior with a minimum in MFPT less than one cell-cycle at an intermediate value of $k_{off}$ (Fig 2D blue and red curves). As the auto-regulatory strength, $\alpha$ is made smaller (stronger repression) the value of $k_{off}$ at the minimum also gets smaller. When repression is "complete" ($\alpha = 0$), i.e. the transcription rate is 0 when bound by repressor, the intermediate minimum disappears (Fig 2D, yellow curve). In Fig 2E, we once again see that the MFPT is collapsed by plotting MFPT as a function of TF occupancy.

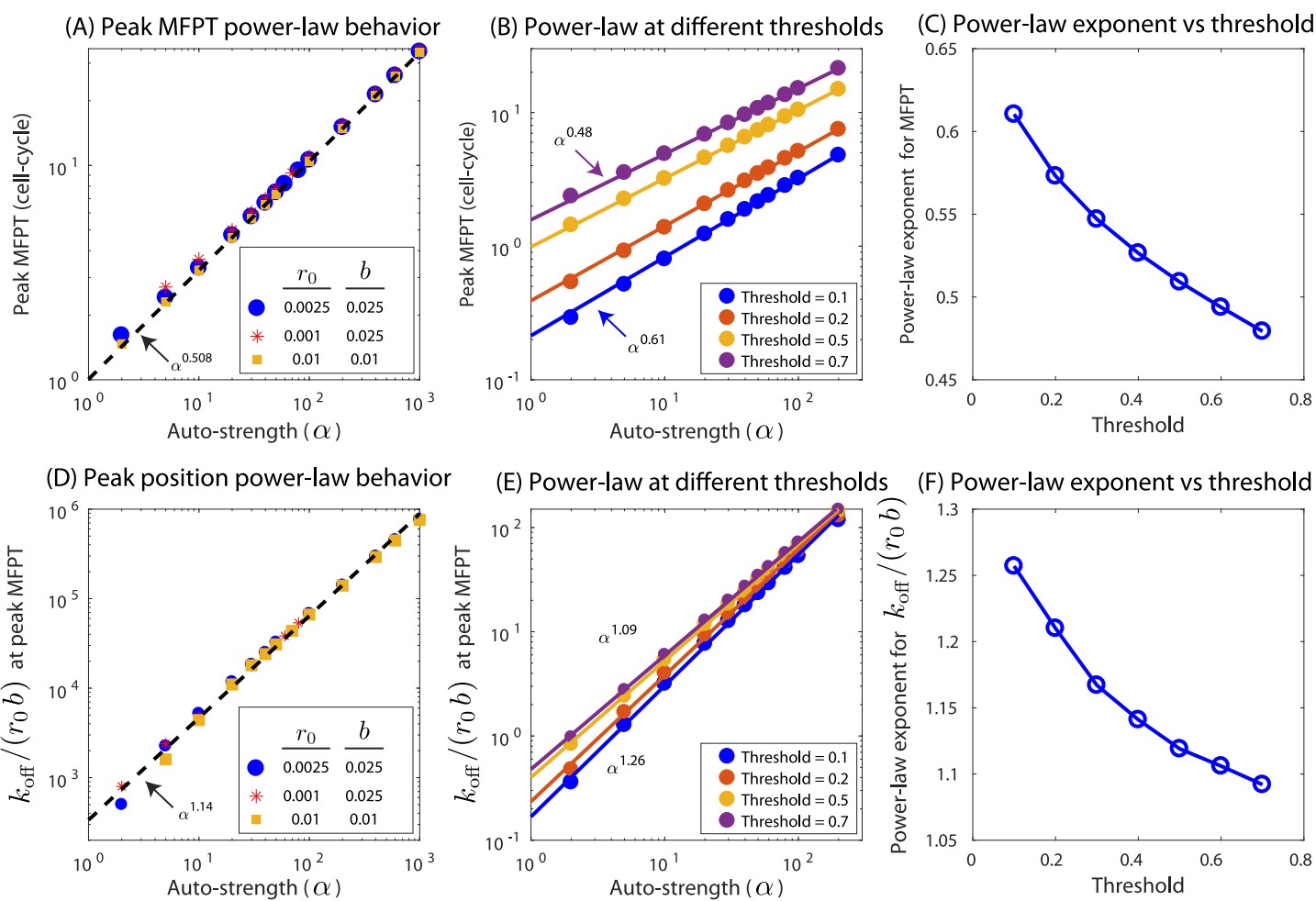

**Fig 3. Power-law behavior in auto-activated gene.** (A) Peak value of MFPT as a function of auto-regulation strength, $\alpha$, shows a power-law behavior with approximate exponent of 0.5 irrespective of basal transcription ($r_0$) or translation rates ($b$). (B) Peak MFPT for different expression thresholds have different exponents. (C)Power-law exponent of Peak MFPT versus auto-regulation strengths as a function of expression threshold. (D) The value of $k_{off}/(r_0 b)$ at the peak MFPT also follows a power law behavior with exponent slightly larger than 1. (E) The exponent again depends on threshold. (F) The value of the exponent as a function of threshold.

We also see, like in auto-activation, the quantity SSE/*R* captures the non-monotonic behavior of the MFPT of an auto-repressing gene as well (Fig 2F).

For an auto-activating gene, we can also examine how the magnitude of the peak in MFPT and the corresponding off-rate at the peak depends on the strength of auto-activation. We find that the peak in MFPT has a power-law dependence on auto-regulatory strength $\alpha$ and the power-law exponent is independent of the transcription and translation rates (see Fig 3A). In this case, the exponent to the power-law is approximately 0.5, meaning for example that a 4-fold increase in the strength of activation will net a 2-fold increase in the peak MFPT. However, the arbitrary choice for MFPT threshold affects the power-law exponent in a monotonic way with lower thresholds having a slightly higher exponent and vice versa (Fig 3B and 3C). Furthermore, the off-rate ($k_{off}$) corresponding to the MFPT peak collapses to a single power-law curve when scaled with basal expression rate, $r_0 b$ (see Fig 3D). In this case, the power-law exponent is approximately 1, and as in our example above a 4-fold increase in activation strength would have a peak in the MFPT when the normalized off-rate is increased approximately 4-fold. Once again, the power-law exponent depends on the chosen value of the MFPT

threshold and the exponent is higher for lower threshold and decreases monotonically as the threshold is increased (Fig 3E and 3F). Furthermore, the MFPT for a constitutive gene can be estimated from the power-law fit by setting $\alpha = 1$. We plot the estimated MFPT and the MFPT computed from simulation for varying parameter ranges in S1B Fig and find comparable results. For auto-repressing genes, the minima in MFPT as well as the off-rates corresponding to the minima deviate from this power-law behavior (see S4A and S4B Fig).

## The role of TF competition in controlling expression timing

Next, we investigate the effect of TF sharing with competing binding sites (decoy). Competing binding sites for the TF reduce the availability of free TF (TF not bound to any gene or competing binding sites). As a consequence, both the steady state expression (SSE) and MFPT is affected. As shown in Fig 4A, the presence of competing binding sites in an auto-activating TF gene has effects similar to varying the TF-binding affinity in that the MFPT once again shows non-monotonic behavior. However, in this case the peak value of the MFPT can be

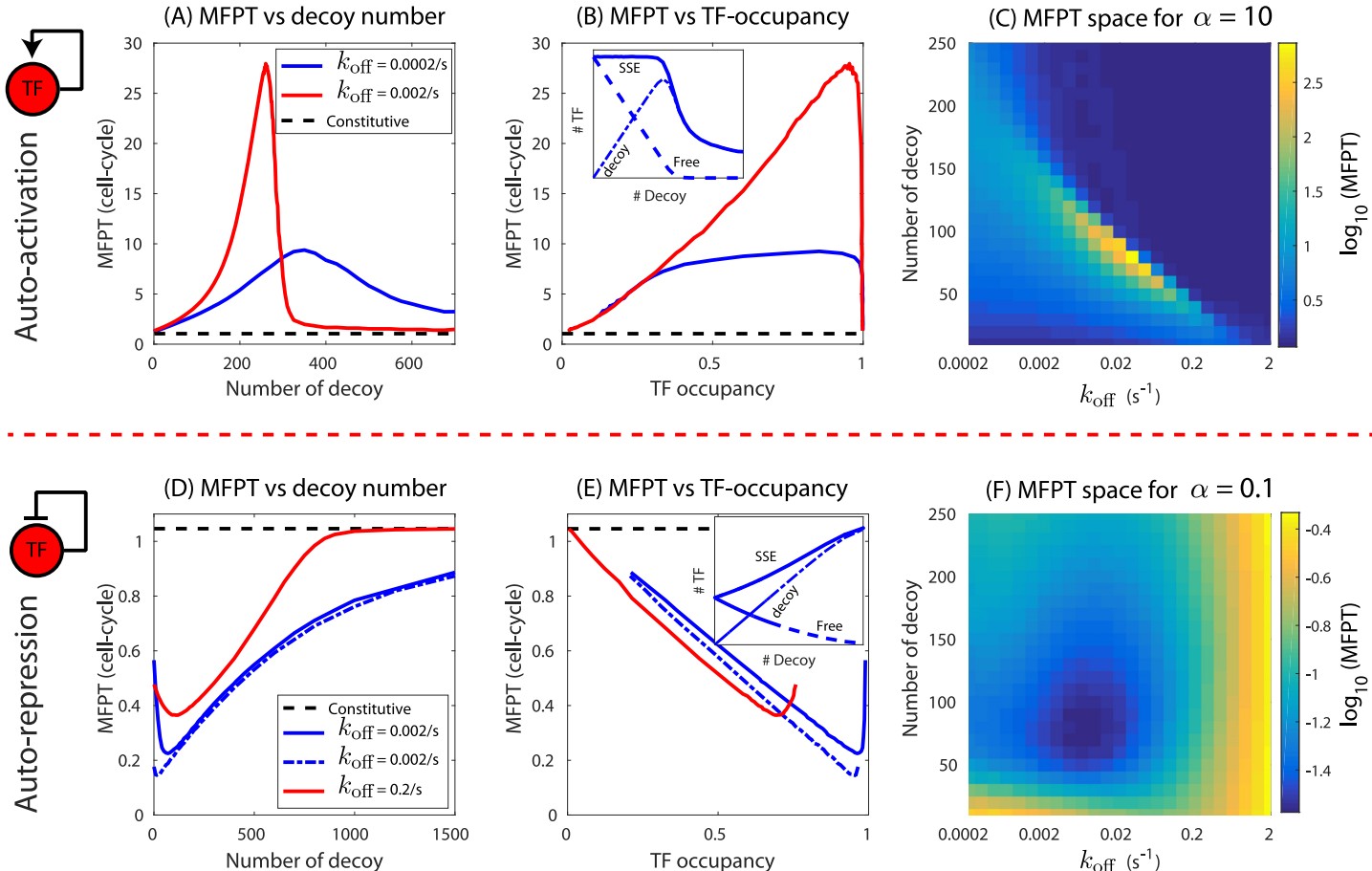

**Fig 4. MFPT of an auto-regulated gene in a network of competing binding sites.** The MFPT of an auto-activated gene as a function of (A) number of competitive binding sites and (B) steady-state occupancy of TF at the promoter when decoy number is varied. The red curve is for weaker TF binding while the blue curve is for stronger TF binding. Inset: MFPT as function of free TF in steady state. Black dashed lines correspond to MFPT of a constitutive gene of transcription rate $r_0$. (C) Heatmap showing phase space of MFPT in number of competing binding sites and affinity space for auto-activating gene ($\alpha = 10$). (D,E) The MFPT of an auto-repressed gene as a function of competing binding sites and steady-state occupancy. Dashed blue line correspond to complete repression ($\alpha = 0$). Inset: MFPT as function free TF in steady state. (F) Heatmap showing phase space of MFPT in number of competing binding sites and affinity space for auto-repressing gene ($\alpha = 0.1$). We use $\alpha = 10$, $b = 0.05 \text{ s}^{-1}\text{mRNA}^{-1}$, $r_0 = 0.0025 \text{ s}^{-1}$ (activation) and $\alpha = 0.1$, $b = 0.05 \text{ s}^{-1}\text{mRNA}^{-1}$, $r_0 = 0.05 \text{ s}^{-1}$ (repression). $\gamma = 0.0003 \text{ s}^{-1}$ is used for protein degradation corresponding to $\tau = 38.5$ min, all additional parameter values listed in the Materials and methods.

significantly higher. The magnitude of the maxima depends on TF binding affinity with weaker sites displaying higher peaks (Fig 4A red and blue curves). The red curve in Fig 4A has an affinity chosen to be consistent with the strong LacI binding site *lacIO1* [40] and we see a predicted maximum in MFPT of 29 cell cycles. The maximum in MFPT becomes higher for weaker binding sites and can reach hundreds of cell-cycles for sites with affinities consistent with *lacO2* ($k_{\text{off}} \sim 0.01 s^{-1}$) or *lacO3* ($k_{\text{off}} \sim 0.6 s^{-1}$). This can be seen in Fig 4C, where we plot a heatmap of MFPT in number of competing binding sites and affinity space. These timescales are exceptionally long compared to, for instance, the total duration of log phase growth of *E. coli* in lab conditions which is typically less than 10 generations. This implies that for specific "network sizes", the TF gene could not reach a significant fraction of its SSE level in a realistic time frame. The peaked MPFT behavior start to disappear as the affinity is further reduced (Fig 4C).

To understand the non-monotonic behavior of the MFPT as a function of decoy site number, we plot the steady state expression (SSE) level, the number of free TFs and the number of TF-bound decoy sites in Fig 4B inset. Once again we will argue based on the balance between the SSE level (a fraction of which must be achieved to achieve the FPT) and the rate at which it gets there, which is a function of the number of free TFs (*i.e.* TFs not already bound to binding sites). When the decoy number is small (less than the constitutive expression level of roughly 390 proteins for this example), the presence of decoy sites do not alter the SSE significantly (solid blue line). However, in this regime, the free TF number reduces linearly with decoy number as the TFs bind to the available decoy sites (dashed line in Fig 4B inset). This reduction in free TF will slow the production rate of TF and the MFPT will increase with decoy number in this regime. This decrease in free TF stops when the number of decoys is greater than the constitutive expression level, at this stage, very few if any TFs are free and the production rate will be roughly insensitive to decoy number. The SSE level in this regime will now decrease rapidly with decoy number signifying an overall decrease in the MFPT. The intermediate maxima will then occur approximately when the number of decoys and SSE levels of TF are comparable. It can also be seen from the heatmap (Fig 4C) and Fig 4A that that the number of decoys corresponding to the peak MFPT reduces as the affinity is lowered; low affinity also corresponds to lower SSE level for TFs. Unlike when affinity is varied, we do not find power-law behavior for the peak MFPT as a function of regulatory strength when decoys are varied (see S4C and S4D Fig).

For a self-repressing gene, we find that with increasing decoy sites, the MFPT first decreases, attains a minimum value and then again goes up asymptotically reaching the MFPT for a constitutive gene with basal rate $r_0$ (Fig 4D). In Fig 4D, we see that at zero occupancy (achieved at high decoy numbers) the MFPT is approximately equal to the MFPT of a constitutive gene (black dashed line). Decreasing the decoy number increases the occupancy down to a minimum which is always less than one and is set by the TF-binding affinity and production rates. The MFPT at this point depends on TF-binding affinity which was shown in Figs 2D and 4D. For complete auto-repression (Fig 4D, dashed blue curve) the MFPT at zero decoy is close to 0.21 cell cycle consistent with [30]. Interestingly, even for complete repression we see a slight dip in MFPT as decoys are introduced. In Fig 4F, we show a heat map of MFPT in number of competing binding sites and affinity space. For $\alpha = 0.1$ the heat map shows a global minimum in MFPT corresponding to $k_{\text{off}} = 0.009 s^{-1}$ and 80 decoys. Moreover, the qualitative behaviors are consistent for different parameter sets (see S2 and S3 Figs, where we have used cell division time of 38.5, 58 and 116 min and mRNA degradation rate of 0.02, 0.01, 0.005 $s^{-1} \text{mRNA}^{-1}$). In addition, the results discussed in the previous section and this section can be reproduced by transforming the stochastic reactions to a deterministic ordinary differential equations (see S2 Text, Fig 1 in S2 Text, S7 Fig, and Fig 1 in S3 Text). Also, we would

like to emphasize that the choice of model for protein degradation, whether protein is diluted purely through cell division or random degradation with a half-life of $ln(2)/\tau$, does not alter our findings (see S1 Text and Fig 1 in S1 Text).

## The precision of expression timing depends on the nature of regulation

Next, we investigate the effect of TF-binding affinity and decoy size in controlling the noise of first passage time. We use coefficient of variation (CV) defined as the ratio of the standard deviation of first passage time and MFPT as a measure of noise. In Fig 5C and 5F, we plot CV of first passage time versus MFPT for auto-activation or auto-repression if either the TF binding affinity or the number of decoys is used as a control parameter of the system. We know from Fig 2 that for auto-activation the MFPT will first rise and then fall as the affinity is increased, Fig 5C shows that despite degeneracy in MFPT (two different $k_{off}$ values give the same MFPT), the associated noise in the MFPT is different for strong and weak binding sites with stronger binding leading to tighter distributions around the mean (Fig 5A). For auto-repression (Fig 5B and 5C) the story is similar but reversed; the MFPT is now less than 1 and

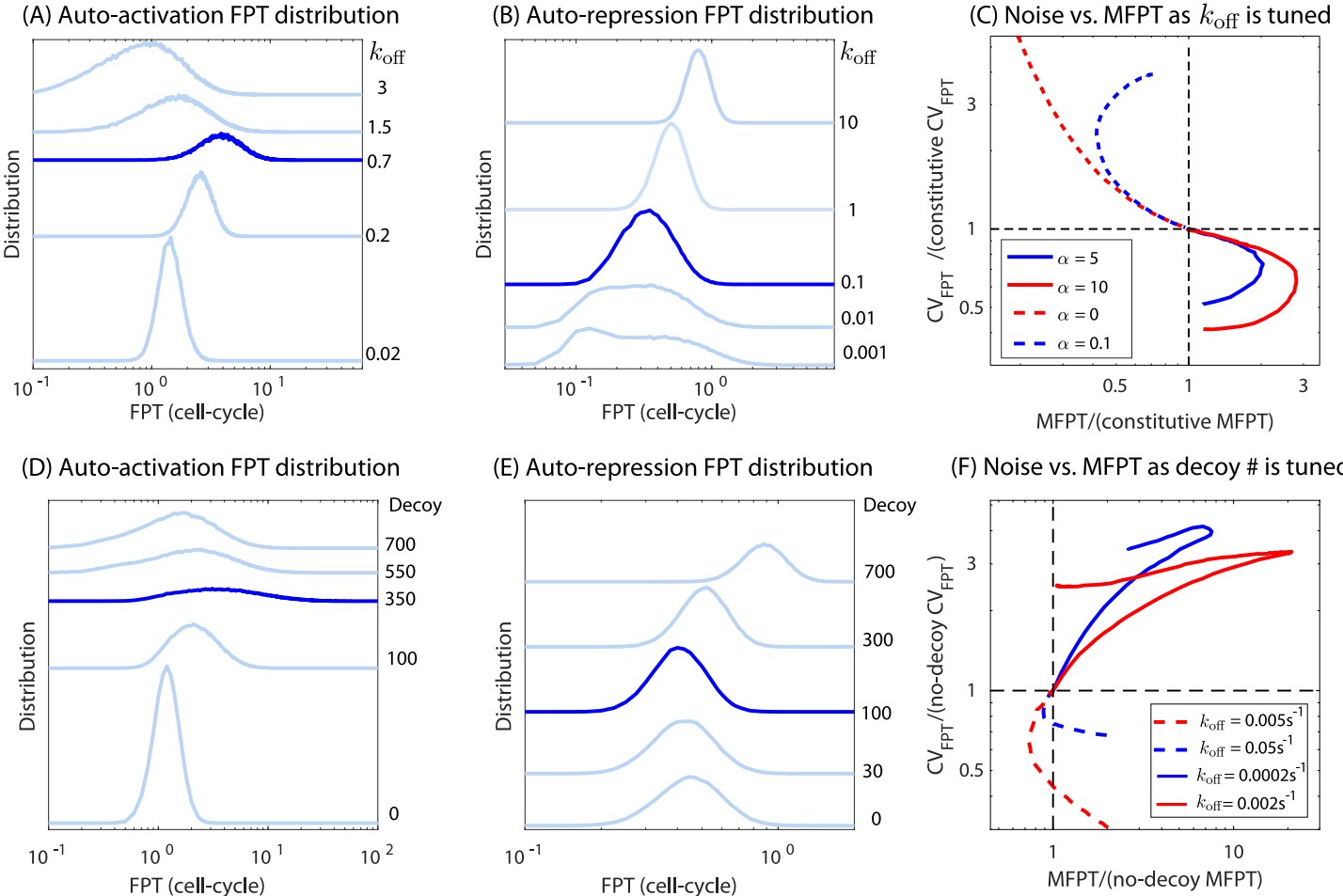

**Fig 5. Distribution of first passage time (FPT) of an auto-activated TF gene.** (A-B) FPT distribution for several values of $k_{off}$ for (A) auto-activation and (B) auto-repression. Dark blue curves represent distribution corresponding to $k_{off}$ where the MFPT peaks or have a minimum. (C) The CV of first passage time as a function of MFPT when $k_{off}$ is changed to vary MFPT for auto-activation (solid lines) and auto-repression (dashed lines) of differing regulatory strength. Parts (D-F) show the same information but for varying decoy instead.

the noise in MFPT is lower for weak binding compared to strong binding site having same MFPT. When number of decoys is used as a control parameter, the CV versus MFPT has trend qualitatively similar to affinity as the control parameter (Fig 5F). In this case, zero competition (no decoy) leads to the minimum CV for auto-activation and maximum CV for auto-repression (Fig 5D and 5E). In Fig 5C, the CV and MFPT are scaled by the CV and MFPT of a constitutive gene. This gives us additional information about the noise associated with an auto-regulated gene. We find that although the mean response time (MFPT) of an auto-activated gene is slower than for an auto-repressed gene, the noise in the response is always lower; the noise of a constitutive gene is the bridge between the two modes of auto-regulation. Also, the noise monotonically increases (auto-activation) or decreases (auto-repression) with $k_{\text{off}}$, which means the noise at peak MFPT is always at an intermediate level. On the contrary, when decoy number is tuned, the auto-activated genes becomes more noisy compared to auto-repressed genes; the CV and MFPT are scaled by the CV and MFPT of the corresponding no decoy network. Interestingly, for an auto-repressing gene having very strong binding affinity, the FPT distribution is bimodal (Fig 5B, bottom curve) and disappears for lower affinities. In all other scenarios, the distribution is unimodal.

## The expression timing of a target gene depends on the nature of TF regulation

In this section, we explore the impact of a TF's mode of auto-regulation on the expression timing of a target gene. The TF gene can be expressed in three different ways: auto-activated ($\alpha >$ 1), auto-repressed ($\alpha < 1$), and constitutively expressed ($\alpha = 1$). In Fig 6A and 6B, we plot the MFPT of an activated (Fig 6A) or repressed (Fig 6B) target gene as a function of TF off-rate at the target promoter ($k_{\text{off,t}}$). The colored lines in the plot represent different modes of auto-regulation of the TF (different values of $\alpha$). In this plot, the number of total TFs is kept constant by adjusting the translation rate of the TF gene to counteract the effect of changing $k_{\text{off,t}}$. Tuning the off rate of the target changes the free TF availability. For instance, if the TF binds to the target gene weakly, more TFs will be available to regulate the TF gene itself and vice versa. This effect is more pronounced at low TF copy number [40].

As the TF-target affinity is decreased the MFPT increases irrespective of the TF auto-regulatory mode or the type of regulation of the target gene (activation, Fig 6A or repression, Fig 6B). For target repression, a lower TF-target affinity corresponds to higher target expression and thus a higher MFPT. On the other hand for target-activation a lower TF-target affinity reduces the target expression but at the same time the rate of production of target is also hampered due to lowering of the affinity which is more substantial. As a result the MFPT of the target gene increases with lowering affinity in the case of activation. One interesting feature revealed from Fig 6A and 6B is that the MFPT curves for different auto-regulatory strength of the TF gene ($\alpha$) never intersect each other. The MFPT curve of a target gene repressed by an auto-repressed TF gene is always greater than the MFPT of a target gene under the control of an auto-activated TF gene and the constitutive TF gene defines the boundary of separation between the two cases (Fig 6B). We get the opposite behavior when the target gene is activated by a TF gene. In this case, the MFPT curve of target gene regulated by an auto-repressed TF gene is always lower than the MFPT of target gene under the control of an auto-activated TF gene with the constitutive TF gene defining the separation between the two (Fig 6A). This tells us that timing of expression of the TF-gene is the key factor for determining the timing of expression of target gene under control of the TF gene.

To further explore this relationship, in Fig 6C, we plot the MFPT of a target gene as a function of the MFPT of the TF-gene which regulates it (where MFPTs are scaled by the respective

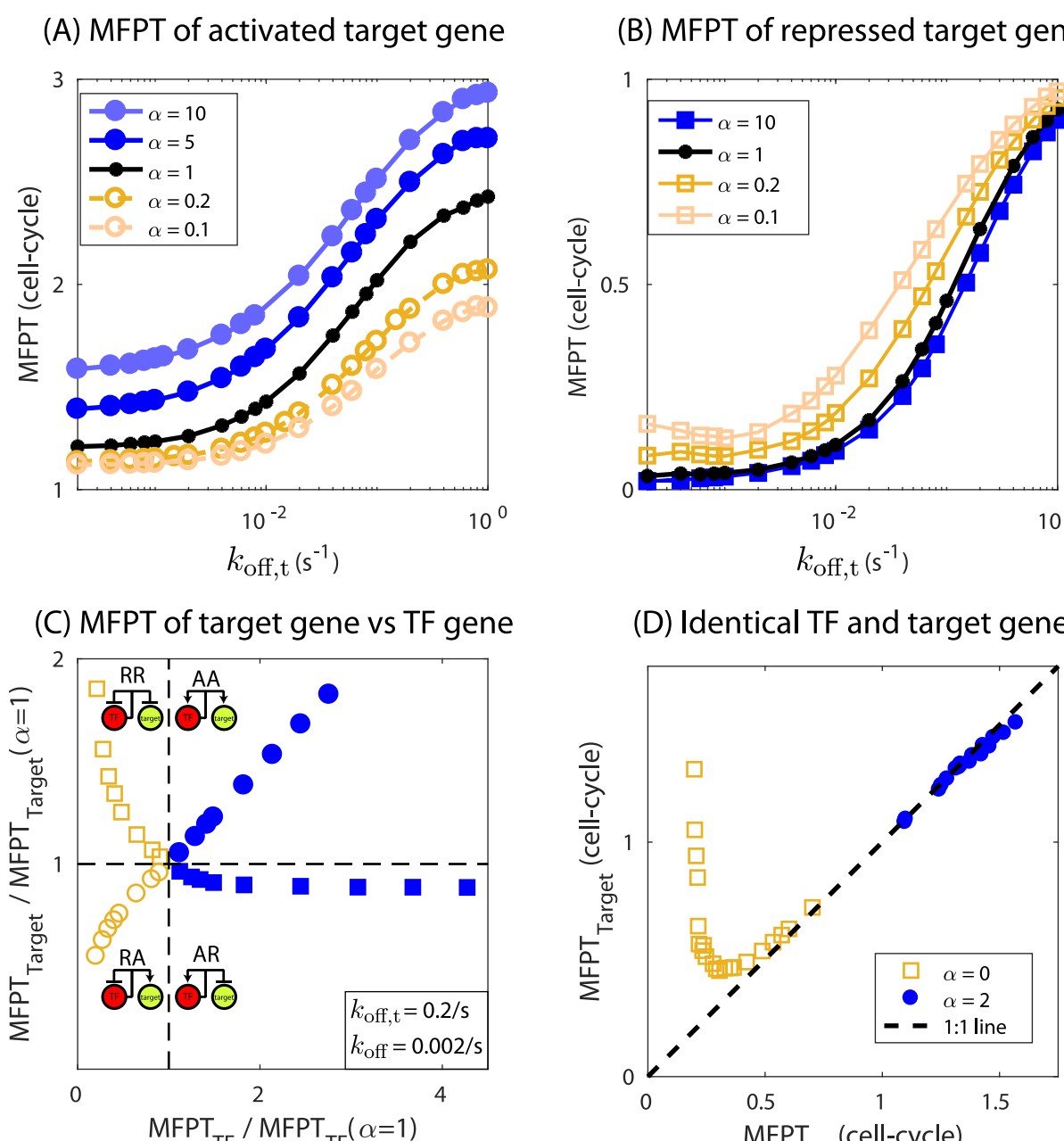

**Fig 6. Expression timing of target gene is dictated by the nature of TF regulation.** (A, B) MFPT of a target gene as a function of binding affinity of TF to the target gene ($k_{off,t}$). Filled symbols represent the TF being auto-activated ($\alpha > 1$), open symbols represent when the TF gene is auto-repressed ($\alpha < 1$), and black filled circles are when the TF gene is constitutive ($\alpha = 1$). Circles represent target gene activation whereas squares represent target gene being repressed by the TF gene. (C) MFPT of target gene as a function of MFPT of TF gene. $\alpha$ is varied from from 0 to 50 while keeping $k_{off} = 0.002$ s$^{-1}$ and $k_{off,t} = 0.2$ s$^{-1}$ constant. The MFPTs of TF and target genes are normalized by their respective MFPTs when $\alpha$ is one or equivalently when TF gene is constitutive. Translation rate of the TF gene is adjusted to achieve constant level of TF number ($\sim 50$) for varying binding affinities and $\alpha$ in (A-C). The target gene is purely activated or repressed, i.e., when a TF is bound to target gene it is expressed in the case of activation and is completely repressed in the case of repression. Parameter used for target gene: $r_{0,t} = 0$, $r_t = 0.05$ s$^{-1}$, $b = 0.1$ s$^{-1}$mRNA$^{-1}$ (target activation), $r_{0,t} = 0.05$ s$^{-1}$, $r_t = 0$, $b = 0.1$ s$^{-1}$mRNA$^{-1}$ (target repression). (D) Plot showing asymmetry in the expression timing of TF and target gene when they have identical binding affinity, transcription and translation rates. Target gene repressed by an auto-repressed TF gene always has higher MFPT compared to the TF gene. TF binding affinity is tuned to generate the curves. Target gene activated by an auto-activated TF gene, on contrary, do not show any significant differences (target gene being moderately faster than the TF gene). Parameters: $r_0 = r_{0,t} = 0.025$ s$^{-1}$, $b = 0.1$ s$^{-1}$mRNA$^{-1}$. Binding affinity is varied to generate the curve.

MFPT for constitutive TF expression, $\alpha = 1$). We find that for target activation, the MFPT of the TF and target gene are positively correlated whereas for target repression it is anti-correlated. Furthermore, MFPT-MFPT phase space is separated into four quadrants; 1) self-repressed gene repressing a target (RR), 2) self-repressed gene activating a target (RA), 3) self-activated gene repressing a target (AR), and 4) self-activated gene activating a target (AA). We find this feature is robust to choice of parameters, see S5A Fig. Interestingly, the two top quadrants, where the MFPT of the target increases are for cases where the TF gene and target have the same mode of regulation (AA or RR). Similarly, the bottom two quadrants, where the MFPT of the target decreases occurs when the TF and target gene are "cross-regulated" (AR or RA). In other words, target genes that are regulated in the same way as their controlling TF have their response slowed, while targets with different regulatory modes than their TFs are accelerated. The histogram in Fig 1D shows that both similarly regulated targets and cross-regulated targets are typical in the networks of autoregulated TFs in *E. coli*. In fact, many autoregulating TFs have both types of target in their networks.

We further explore the expression timing relationship when the binding sequence for TF and target genes are exactly identical, which means the binding affinity of TFs being identical for both TF and target gene. In our previous work [40], we have shown that an auto-repressed gene and its target gene have different steady state expression levels even though their binding sequences are exactly identical. In Fig 6D, we plot the MFPTs of the target gene which is regulated by an auto-regulated TF gene versus MFPT of the TF gene, the regulation being identical for both TF and target gene. Each curve is generated by tuning the TF affinity for TF and target gene while keeping other parameters constant. We find that the target gene is always slower than the TF gene to reach half of their respective steady state expression for negative regulation, i.e., target gene repressed by an auto-repressed TF gene (Fig 6D, open squares. Also, see S5B Fig, where number of decoy sites are varied keeping the affinity fixed). The difference in their timing is maximum for high affinity binding sites and approaches to one-to-one line showing similar timing in expression as the affinity is lowered. For target gene activated by an auto-activated TF gene, we find that the target gene is faster than the TF gene, however, the difference is not significant (Fig 6D, filled circles). The noise of the target gene has a complex relationship with the noise of the TF gene and depends on the model parameters. In the case of target activation, usually we find that the noise of the target gene is always less than that of the TF gene. However, when the target gene is repressed, the noise can be higher or lower compared to the noise of TF gene (see S8 Fig).

## Discussion and conclusions

The decay rate of proteins sets the natural timescale of the response of transcriptional networks [21]; in bacteria where proteins are long-lived this timescale tends to be the division rate of the organism. Although specific network connections are known to speed or slow this response by several fold, extremely slow or fast responses have been observed in a growing number of systems [22–25]. Here, we systematically investigate the regulatory features that control gene expression timing of auto-regulated TFs, specifically in determining the mean first-passage time (MFPT) of reaching half of the steady-state level of expression.

Interestingly, we find that auto-activation and auto-repression, around a specific range of TF binding affinities, are capable of slowing or speeding response times by orders of magnitude. Similarly, we find that extra TF binding sites can alter the timing response in a more dramatic fashion. These extra binding sites may serve to regulate other promoters in the network or they may serve no specific function (*i.e.* decoys). We highlight the tremendous range of responses that are achievable from the auto-regulatory architecture; specific values of binding

affinity and network size are capable of tuning the natural timescale of response for an auto-regulated gene over most timescales that would be relevant for cellular response. In the extreme cases, a gene may not be realistically able to reach steady-state levels in physiologically reasonable timescales, requiring hundreds of generations simply to reach half of the steady-state level. Despite arguments of the inherent interest of "steady-state levels" in such networks the concept is essentially irrelevant.

One curious aspect of the response time modulation in this regulatory architecture is that it occurs at intermediate parameter values, rather than at the extremes. We find that this peak behavior arises due to an interplay between modulation of the production rate and modulation of the steady state level; the ratio of these two quantities has a maximum (for activation) or a minimum (for repression) at intermediate levels of both TF unbinding rate and network size. Furthermore, both the location and magnitude of this peak scales with the "strength" of the auto-activation with a well defined power law. The mechanisms for generating power law behavior are typically associated with stochastic processes [58], however in this case the emergence of this property can be seen even in the deterministic treatment of the system. Despite not identifying the root mechanism behind this, the power law type behavior enables a simple intuition about how the dynamics will change with the regulatory strength of the TF (*i.e.* making it a better or worse activator); a TF that is a four times stronger activator will have a maximum MFPT that is doubled.

We also examine how the fidelity of single-cell timing depends on these physical parameters. Due to the intermediate extrema in MFPT, there are typically two values of TF binding affinity or decoy number that produce an identical "mean response", however these two regulatory parameter sets produce significantly different variance in timing among individual cells. As a result, for the majority of achievable MFPTs of an auto-regulatory gene there is both a "quiet" and "noisy" way to achieve a given response time. For auto-activation, higher off-rates and larger numbers of decoys produce noisier responses while the opposite is true for auto-repression. Just like for the case of average gene expression levels where it is argued that the ability to tune noise independent of mean can be important for fitness, it stands to reason the same holds for the speed of response of a gene's transcriptional response and thus the ability to reach similar average response times with very different levels of noise can be useful.

Finally, we show how the response of a target gene is controlled by the mode and details of regulation of the TF gene. Interestingly, if the mode of regulation for both the TF and target are the same (*i.e.* both repressed or both activated) the MFPT of the target will increase, whereas a "cross-regulated" target (one where the TF gene and target gene are differently regulated) will be sped up. Both of these types of regulation, similarly regulated and cross-regulated, are common in *E. coli*; many auto-regulated TFs having both activated and repressed targets (see Fig 1D).

Overall, understanding the molecular features that dictate the timing in expression for genes is an important part of reading natural gene networks and designing efficient expression timing for synthetic networks.

## Materials and methods

### Model and simulation methodology

Here, we describe a generalized model for an auto-regulatory TF-gene, one target gene and multiple competing binding sites. The TF-gene product, TF protein (X), binds to its own promoter ($P_{\text{TF}}$), to the promoter of the target gene ($P_{\text{Target}}$), and to the decoy sites ($N$) with a constant rate $k_{\text{on}}$ per free TF per unit time. TF unbinds from the promoter with a sequence

dependent off-rate ($k_{off}$) which determines the TF-binding affinity of the binding site. The off-rates of a bound TF from the promoters of the TF and target, and from the decoy sites are $k_{off}$, $k_{off,t}$, and $k_{off,d}$ per unit time, respectively. mRNAs are produced at the rate $r_0$ from a TF-free promoter (basal transcription) and rate $r$ from a TF-bound promoter. The ratio of $r_0$ and $r$ determines whether the gene is repressed ($r/r_0 < 1$) or activated ($r/r_0 > 1$). This ratio is termed auto-regulatory strength, $\alpha$. Furthermore, $\alpha = 1$ describes a constitutive gene. mRNAs are translated into protein at the rate $b$. The mRNAs and the proteins are degraded at the rate $\gamma_m$ and $\gamma$, respectively. The lifetime of protein in *E. coli* and many other organisms are much longer, typically 2–3 hrs, and hence the protein degradation, $\gamma = ln(2)/\tau$, is assumed to be due to dilution during cell division ($\tau$). We further assume that all proteins (free protein, TF bound to promoter and TF bound to decoy sites) degrades with the same rate. The set of reactions describing the model above are listed below.

$$m_{TF(Target)} \xrightarrow{r_0} m_{TF(Target)} + 1 \quad \text{(Production of mRNA from TF free promoter)}$$

$$m_{TF(Target)} \xrightarrow{r} m_{TF(Target)} + 1 \quad \text{(Production of mRNA from TF bound promoter)}$$

$$m_{TF} \xrightarrow{b} m_{TF} + X \quad \text{(Translation of TF protein)}$$

$$m_{Target} \xrightarrow{b} m_{Target} + Y \quad \text{(Translation of target protein)}$$

$$X + P_{TF(Target)} \xrightarrow{k_{on}} XP_{TF(Target)} \quad \text{(Binding of a TF to a promoter)}$$

$$XP_{TF(Target)} \xrightarrow{k_{off,TF(Target)}} X + P_{TF(Target)} \quad \text{(Unbinding of a TF from a promoter)}$$

$$m_{TF} \xrightarrow{\gamma_m} m_{TF} - 1 \quad \text{(Degradation of mRNA)}$$

$$m_{TF(Target)} \xrightarrow{\gamma_m} m_{TF(Target)} - 1 \quad \text{(Degradation of mRNA)}$$

$$X + N \xrightarrow{k_{on}} XN \quad \text{(Binding of a TF to decoy)}$$

$$XN \xrightarrow{k_{off,Decoy}} X + N \quad \text{(Unbinding of a TF from decoy)}$$

$$X, Y \xrightarrow{\gamma} \phi \quad \text{(Degradation of TF protein)}$$

$$XP_{TF(Target)} \xrightarrow{\gamma} P_{TF(Target)} \quad \text{(Degradation of TF from promoter} - \text{complex)}$$

$$XN \xrightarrow{\gamma} N \quad \text{(Degradation of TF bound to Decoy)}$$

We implement the simulations for stochastic reaction systems using Gillespie's algorithm [44] in C programming, to compute mean first passage time (MFPT). The threshold for MPFT is set to 50% of steady state expression (SSE) unless specified. Simulations are run for sufficiently long time ($\sim 10^6$ s) to get rid of the transient dynamics and then steady state distributions (TF and target protein expressions as well as the promoter occupancy) are obtained by sampling over time with a time interval ($T_S$), long enough for the slowest reaction to occur 20 times on average ($T_S = 20$ over rate for slowest reaction). We also, record whether the promoter is occupied or not and assign a value 1 (occupied) or 0 (unoccupied) for each sampling. The average of this value gives the TF-occupancy. Next, we run the simulations using the steady state expressions as input to obtain gene expression trajectories. At time $t = 0$, the

network is switched on to express until the expression reaches a threshold. The time for each trajectory to reach that threshold is termed as first passage time and their mean is referred as mean first passage time (MFPT). MFPT and noise in first passage times are computed using at least $10^5$ individual trajectories. For simulations, we use $k_{\text{on}} = 0.0027$ s$^{-1}$ ([51]), $\gamma_{\text{m}} = 0.011$ s$^{-1}$ ([59]) and $\gamma = 0.0001, 0.0002, 0.0003$ s$^{-1}$. These rates correspond to 90 sec average life for mRNA and a cell-cycle of 115, 56 and 38.5 min, respectively. $k_{\text{off}}$ is varied in a range of 0.0002 − 100 per TF per second to cover the binding affinities corresponding to a wide varieties of natural and synthetic promoters in *E. coli* such as *lacOID, lacO1, lacO2, lacO3*. Typical transcription rate in *E. coli* is of the order of 0.33 s$^{-1}$ for constitutive gene [60], as such we use a basal rate in a range $r_0 = 0.001 − 0.05$ s$^{-1}$ such that the ten -fold activation correspond to a transcription rate of $r = 0.01 − 0.5$ s$^{-1}$. Protein numbers in *E. coli* varies from few units to thousands [61] as such we vary translation rate (*b*) in a range of $0.025 − 0.1$ s$^{-1}$mRNA$^{-1}$. Since, the target proteins do not feed back, the transcription and translation rates of target protein have no impact on the MFPT of target gene. For Fig 6A–6C, we have assumed the target is purely activated (expressed when TF is bound) or purely repressed (no expression when TF is bound). However, for Fig 6C, where the TF and target gene is identical, we have used the same parameters for both TF and target gene. Individuals parameters for target gene is mentioned in the figure captions.

In our model, we do not consider the effects of gene replication explicitly. Depending on the location, the TF gene and the target gene can have different copy numbers at different time point of a cell-cycle. This copy number variation can play a crucial role in determining both the response time and the noise. We also assume that the competing sites are identical, i.e., the TFs bind to the competing sites with the same on and off rate or equivalently the same binding affinity. However, in practice the binding affinity of different sites can be very different. Furthermore, we assume that the TF concentration is uniform throughout the cell the consequence of which is that the TF binding rate is constant and same for all the binding sites. One of the recent studies highlighted that the TF concentration is determined by the gene location and can be non-homogeneous [62].

## Supporting information

**S1 Text. Cell-cycle effect.**
(PDF)

**S2 Text. Deterministic approach.**
(PDF)

**S3 Text. Toy-model for power-law behavior.**
(PDF)

**S1 Fig. Effect of production rates on MFPT.** (A) Effect of transcription and translation rates on MFPT of a constitutive gene. At low transcription rates the MFPT of constitutive gene is usually higher than that at higher transcription rate. Inset: The translation rate on the other hand does not alter MFPT significantly. (B) Comparison of MFPT of a constitutive gene computed from simulation and predicted from the the power-law fit of peak MFPT versus $\alpha$ by setting $\alpha = 1$. For the power-law fit data point corresponding to $\alpha = 1$ is excluded, since $\alpha = 1$ is a constitutive gene having constant MFPT as a function of $k_{\text{off}}$.
(EPS)

**S2 Fig. Effect of protein degradation rate.** Effect of TF-binding affinity and competing binding sites on MFPT for auto-regulated genes for different cell division rates, $\tau = 38.5$ min

(yellow), 58 min (red) and 116 min (blue). The TF gene is auto-activated with (A,C) or auto-repressed (B,D). Dashed lines correspond to MFPT of a constitutive gene of transcription rate $r_0$. Parameters used to generate the figure: $\alpha = 10$, $r_0 = 0.0025$ s$^{-1}$, $b = 0.025$ s$^{-1}$mRNA$^{-1}$ (auto-activation). $\alpha = 0.1$, $r_0 = 0.05$ s$^{-1}$, $b = 0.05$ s$^{-1}$mRNA$^{-1}$ (auto-repression). $k_{\text{off}} = 0.002$ s$^{-1}$ is used for (A,B).
(EPS)

**S3 Fig. Effect of mRNA degradation rate.** Effect of TF-binding affinity and competing binding sites on MFPT for auto-regulated genes for different mRNA degradation rates, $\gamma_{\text{m}} = 0.02$ (yellow), 0.01 (red) and 0.005 (blue) s$^{-1}$mRNA$^{-1}$. The TF gene is auto-activated with (A,C) or auto-repressed (B,D). Dashed lines correspond to MFPT of a constitutive gene of transcription rate $r_0$. Parameters used to generate the figure: $\alpha = 10$, $r_0 = 0.0025$ s$^{-1}$, $b = 0.025$ s$^{-1}$mRNA$^{-1}$ (auto-activation). $\alpha = 0.1$, $r_0 = 0.05$ s$^{-1}$, $b = 0.05$ s$^{-1}$mRNA$^{-1}$ (auto-repression). $k_{\text{off}} = 0.002$ s$^{-1}$ is used for (A,B).
(EPS)

**S4 Fig. Power-law behavior.** (A,B) Power-law behavior in self-repressing gene when affinity is varied. For an auto-repressing gene both the peak MFPT (A) and $k_{\text{off}}$ at the peak (B) deviate from the power-law fit as a function of auto-regulatory strength $\alpha$. Black dashed lines corresponds to power-law fit for auto-activating TF gene (i.e., $\alpha > 1$). Open circles are auto-repression ($\alpha < 1$) and filled circles are auto-activation ($\alpha > 1$). (C,D) Peak MFPT and number of decoys at the peak versus regulatory strength ($\alpha$) when decoy is tuned. We do not observe any power-law behavior. Parameters: $k_{\text{off}} = 0.002$ s$^{-1}$, $r_0 = 0.001$ s$^{-1}$, $b = 0.01$ s$^{-1}$mRNA$^{-1}$.
(EPS)

**S5 Fig. Expression timing of target gene depends on the nature of TF regulation.** (A) MFPT of target gene as a function of MFPT of TF gene. $\alpha$ is varied from from 0 to 50 while keeping $k_{\text{off}} = 0.002$ s$^{-1}$ and $k_{\text{off,t}} = 0.0002$ s$^{-1}$, 0.002 s$^{-1}$, 0.2 s$^{-1}$ constant. The MFPTs of TF and target genes are normalized by their respective MFPTs when $\alpha$ is one or equivalently when TF gene is constitutive. Translation rate of the TF gene is adjusted to achieve constant level of TF number ($\sim$ 50). The TF gene is auto-activated (filled symbols) or auto-repressed (open symbols) and the target gene is activated (circles) or repressed (squares). (D) Plot showing asymmetry in the expression timing of TF and target gene when they have identical TF-binding affinity, transcription and translation rates. Decoy binding sites are varied along the curve from 0 to 300 for $\alpha = 2$ (blue) and $\alpha = 0$ (brown).
(EPS)

**S6 Fig. Fast protein degradation: $\tau = 10$ min.** MFPT of an auto-activated (A) and and auto-repressed (B) TF gene as a function of TF off-rate ($k_{\text{off}}$). The dashed lines correspond to MFPT of a constitutive gene. (C) The CV of first passage time as a function of MFPT when koff is changed to vary MFPT for auto-activation (solid lines) and auto-repression (dashed lines) of differing regulatory strength. (D) Peak value of MFPT as a function of auto-regulation strength, $\alpha$, when $k_{\text{off}}$ is tuned (circles). The dashed line is a power-law fit with an exponent 0.48. (Inset) The values of $k_{\text{off}}$ at the peak MFPT as a function of $\alpha$ also follow a power-law with an exponent 1.12. Parameters: $\gamma_{\text{m}} = 0.01$ s$^{-1}$mRNA$^{-1}$, $\alpha = 10$, 20, $r_0 = 0.0025$ s$^{-1}$, $b = 0.025$ s$^{-1}$mRNA$^{-1}$ (auto-activation), $\alpha = 0.1$, 0.5, $r_0 = 0.01$ s$^{-1}$, $b = 0.1$ s$^{-1}$mRNA$^{-1}$ (auto-repression).
(EPS)

**S7 Fig. Power-law behavior for different combinations of degradation rates.** Power-law behavior of the peak MFPT (A) and $k_{\text{off}}$ at the peak position (B) for varying mRNA degradation rates and protein degradation rates. Following pairs of degradation rates ($\gamma_{\text{m}}^{-1}$, $\gamma^{-1}$) are

used: 2 min, 50 min (blue), 20 min, 50 min (red) and 20 min, 10 min (yellow). The productions rates are $r_0 = 0.001$ s$^{-1}$, $b = 0.01$ s$^{-1}$mRNA$^{-1}$. Plots are generated using deterministic model. (EPS)

**S8 Fig. Comparison of noise (CV) of an auto-regulated TF gene and a target gene regulated by the TF gene.** (A) The CV of target gene repressed by the TF gene is usually higher than the TF gene irrespective of the mode of TF production, i.e auto-activated TF (blue filled squares), constitutive TF (black squares) and auto-repressed TF (open yellow squares). On the contrary, the CV of target gene activated by a TF gene is lower than the TF gene irrespective of the mode of TF production, i.e auto-activated TF (blue filled circles), constitutive TF (black circles) and auto-repressed TF (open yellow circles). Off-rate of the TF from the target gene ($k_{off,t}$) is varied to generate the curve keeping $k_{off}$ constant. the parameters are same as Fig 6A and 6B in the main text. (B) TF gene and target gene is exactly identical, all the rates being same, and $k_{off}$ is varied along the curve. CV of a target gene repressed by an auto-repressed TF gene (yellow squares) can be high or low compared to the TF gene and depends on the TF off-rate. CV of a target gene activated by an auto-activated TF gene is always less than the TF gene (blue circles).
(EPS)

# Acknowledgments

We wish to thank Vinuselvi Parisutham, Sunil Guharajan and Sandeep Choubey for helpful discussions.

# Author Contributions

**Conceptualization:** Md Zulfikar Ali, Robert C. Brewster.

**Data curation:** Md Zulfikar Ali.

**Formal analysis:** Md Zulfikar Ali.

**Funding acquisition:** Robert C. Brewster.

**Investigation:** Md Zulfikar Ali.

**Methodology:** Md Zulfikar Ali.

**Project administration:** Robert C. Brewster.

**Resources:** Robert C. Brewster.

**Software:** Md Zulfikar Ali.

**Supervision:** Robert C. Brewster.

**Validation:** Md Zulfikar Ali, Robert C. Brewster.

**Visualization:** Md Zulfikar Ali, Robert C. Brewster.

**Writing – original draft:** Md Zulfikar Ali, Robert C. Brewster.

**Writing – review & editing:** Md Zulfikar Ali, Robert C. Brewster.

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
