## [Decision Letter · Decision Letter 0]

19 Jul 2021

Dear Brewster,

Thank you very much for submitting your manuscript "Controlling gene expression timing through gene regulatory architecture" for consideration at PLOS Computational Biology.

As with all papers reviewed by the journal, your manuscript was reviewed by members of the editorial board and by several independent reviewers. In light of the reviews (below this email), we would like to invite the resubmission of a significantly-revised version that takes into account the reviewers' comments.

We cannot make any decision about publication until we have seen the revised manuscript and your response to the reviewers' comments. Your revised manuscript is also likely to be sent to reviewers for further evaluation.

Sincerely,

Alexandre V. Morozov, Ph.D.

Associate Editor

PLOS Computational Biology

Sushmita Roy

Deputy Editor

PLOS Computational Biology

Reviewer's Responses to Questions

**Comments to the Authors:**

Reviewer #1: Summary:

In “Controlling gene expression timing through gene regulatory architecture”, Ali and Brewster explore qualities of transcription factor (TF) circuits that tune timescales of activation or deactivation of gene expression states. Their work adds to a field that often focuses on steady-state characteristics of gene expression.

Ali and Brewster ask how the parameters of one of the simplest circuit motifs (a single node with feedback) influence the speed at which the circuit reaches steady state expression (SSE). The authors chose a well-precedented framework for the investigation by applying a stochastic simulation algorithm (Gillespie algorithm). This allows them to evaluate single-cell trajectories, getting both fine detail as well as characterizing bulk population dynamics. Using first passage time (FPT) as a metric for the timescale at which a circuit progresses to a new equilibrium, they modulate binding affinity, decoy sites, and auto-strength, and examine the resulting effects on FPT. With these analyses, the authors make several findings that may help guide analysis of native gene circuits and design of gene circuits with desired timing properties.

The dynamics of regulatory circuits is a timely and important problem, particularly in designing more complex circuits that may interact with dynamic signals (Gerardin et al., Cell Systems, 2019; Bashor et al., Science, 2019). In addition to analyzing bulk population timing response with the summary statistic mean FPT (MFPT), Ali and Brewster also use the stochastic simulation to illustrate variance among single cell trajectories. The two analyses together find that MFPT can be achieved with different noise levels, enabling design strategies that can tune both variance and average timing.

Separately, the use of decoys has recently taken focus as a strategy for tuning steady-state expression levels (Wan et al., Nat Commun, 2020; Wang et al., Nucleic Acids Res, 2021), and have played an important role in controlling timing aspects in dynamics systems as well (Potvin-Trottier et al., Nature, 2016; Henningsen et al., ACS Synth Biol, 2020). Here, Ali and Brewster quantitatively evaluate the role of decoys on timing of a single node with feedback. They find an impactful result in that decoys may increase FPT tremendously up to 100s of generations, such that the system might not actually reach SSE.

Finally, the authors examine the effects on target gene regulation alongside auto-regulation. They find a large space of regulatory timing that can be produced by single node feedback. The work indicates that auto-regulatory architecture is fundamental in timing of the target.

Overall, the paper succeeds in exploring the effects of TF circuit parameters on the timescales of the system and its targets. The results inspire new ideas about how to analyze and design TF circuits with desired timing properties. Specific comments and questions that could enhance the work are enumerated below.

Specific questions and comments:

1. There are several directly relevant works that should be cited as state-of-the-art for how the temporal characteristics of gene circuits have been explored and how they are currently designed.

a. Specifically, in “The Design Principles of Biochemical Timers: Circuits that Discriminate between Transient and Sustained Stimulation” by Gerardin, Reddy, and Lim, basic network motifs (including 1-node feedback architectures) are extensively characterized with focus on timing of responses. These characteristics include “trigger time,” which is analogous to FPT, as well as a host of other qualities.

b. Additionally, in “Complex signal processing in synthetic gene circuits using cooperative regulatory assemblies” by Bashor et al., the same circuit parameters (regulatory architecture and binding affinity, i.e., koff) are examined across a large space of configurations to generate a space of “activation/deactivation half-times,” which is again analogous to FPT. In addition to the computational model space examined, the work supports the predictions with synthetic circuits in living cells.

A fundamental difference among the papers mentioned and the present Ali and Brewster work is the kind of model used. The application of a stochastic simulation model in the present work enables different kinds of analysis such as a variation in timing among single cells, “noise”.

2. For the power-law relationship between peak MFPT and auto-strength (alpha) during auto-activation, can the authors comment further on (A) why this power law is generated, (B) why this power law does not exist in auto-repression, and (C) why the exponent changes with MPFT threshold?

3. The authors analyze a large range of decoys that can be used to tune regulatory timing. The plots in Figure 4 look at hundreds to thousands of decoys used, and the authors suggest that at high decoy numbers, cells can generate an interesting phenomenon where they may never reach steady state. What is the context at which these numbers are reasonable to consider? For instance, for a particular native transcription factor, are hundreds to thousands of sites present in a genome? Or if designing synthetic circuits, is it reasonable to deliver such a large payload with high repetitiveness?

4. The few examples of noise vs koff show that it is monotonically increasing (auto-activation) or decreasing (auto-repressing) with increasing koff. It is unclear if this holds for all or a large region of the parameter space explored. Does the koff at the local extrema in MFPT always generate some intermediate noise level?

5. The authors make an interesting comment that the FPT distribution is bimodal in strong auto-repression. Out of curiosity, is there any idea as to why this occurs? Is this largely beyond physiological ranges?

6. In examining expression timing of a target gene, the authors chose to include auto-repressors that activate targets and auto-activators that repress targets. Interestingly, it is in these cases that the target regulation is sped up compared to constitutive targets. How common are these regulatory architectures in nature and is there a precedent for designing them in synthetic contexts?

7. The authors take generation time as the dominant force of decay for proteins. In a stochastic model, how impactful would it be to consider decay as a large synchronous event (i.e., cell division) instead of spontaneous degradation of individual molecules over time (as is currently assumed)?

8. The caption on Figure 5 has mixed up the panel reference. It refers to (C) as (A) and (A-B) as (B-C).

9. Should lines 136-139 say that SSE decreases as binding affinity decreases (i.e., k_off increases)?

10. CV should be defined the first time it is used in the paper in the “Results” section, subheading “Model”, line 91.

11. In supplemental Figure S3, panel (B), the title reads ‘locaiton’ instead of ‘location’. Also in Fig S3, are open circles auto-repressing (alpha < 1) and the caption has it reversed?

Reviewer #2: The authors develop and analyse kinetic Monte Carlo simulations to investigate how the timing of gene expression in auto-regulated genes (activation or repression) is affected by TF binding affinity and competition for TFs (via decoy sites).

Most of the manuscript is a nice reading, with simple concepts and approaches. I am surprised no one thought to investigate auto-regulated circuits with a simple computational approach such as this one (this is not to be intended as a negative comment).

I would suggest the authors to better introduce the literature on the topic, as it is already partly done in the introduction, and in particular emphasise more the analogies, differences and breakthroughs of their works. For instance, what are the differences with, e.g. ref.[27]? To my understanding the most relevant feature introduced in Ali and Brewster is the presence of a limited number of TFs, which allows them to study the effects of the competition with other binding sites.

I understand that taking a threshold that is relative to the steady-state value is something consistent with the literature, but would the results change by taking an absolute value of expression as a threshold? I imagine that in many regulatory systems it is the concentration of TFs determining the expression (or not) of a product, not the relative expression with respect a steady-state value.

I have the impression that sometimes the authors could dig more to rationalise their results. For instance, what is the origin of the power-law dependences in Fig.3? The CV is discussed in a few lines only, and panels (C) and (F) of Figure 5 are difficult to understand and could be introduced better in the text.

An attempt to make an analytical theory on a few aspects could have been tried (even just a few back to the enveloppe estimates).

Furthermore, discussing the biological consequences of the results could help to increase the impact and resonance of this paper. I do not know if there are available datasets that could be analysed, but in case I would encourage the authors to do so.

The manuscript sometimes lacks of precision. For instance it seems to me that the caption in Fig.3 does not correspond to what is represented in the figure (insets), and in the caption of Fig.5 the panels have been inverted. In the same would be helpful to plot the ratio CV/CV(constitutive)? What is the role (and value) of $\\alpha_t$ (introduced in Fig.1) in Figure 6?

The authors should include more details on their method, expanding the dedicated section to facilitate the reproduction of their results (see also the comment on $\\alpha_t$ above). For instance, how is the 50% threshold calculated? Is it the analytical estimate or the value of the (mean) numerical steady-state? Sentences like " In this plot, the number of total TFs is kept constant by adjusting the translation rate to counteract the effect of changing koff" about Figure 6 are not easily understandable without a deeper discussion.

I would suggest to edit and expand the last 2 sections of the paper (from page 9), the explanation and discussion of the results.

MINOR

- In the Author Summary: "which dictate"  "which dictates"?; "pose a problem"  "poses a problem"?

- Top page 2, "TF" already introduced on line 2.

- Line 16: "However, despite the prevalence of this regulatory motif in natural genetic circuits and the importance for efficient transcriptional programs, we can not predict or design temporal gene expression patterns from this motif based on the regulatory sequence of the regulated genes." Why? I am not sure I understand. Can the author explain better?

- "Importantly, the MFPT is a single-cell measurement of response time; as opposed to measuring how long the bulk takes to reach a certain expression level on average." Can the authors further develop this sentence?

- Why only changes in $k_{off}$ (and not on $k_{on}$) are considered? Is this because the binding events can be assumed to be always diffusion limited?

- In the references, 'jacob''Jacob'

Reviewer #3: In this work the authors use stochastic simulations to analyse the timing of gene activation or repression for the “single-input module” network motif, where a TF regulates itself and a target gene. They characterise the mean and the variance of the first passage time (FPT) distribution for the autoregulated TF alone and a target gene, for activating and repressive TF function. They focus on how these quantities depend on the TF unbinding rate (affinity) and the number of competing binding sites on the genome. For the autoregulated gene alone, they find a nonmonotonic relationship between affinity and MFPT and between number of competing sites and MFPT. The relationship between the noise of the timing and these two parameters depends on the type of regulation. Finally, they show that the MFPT of the TF gene determines that of a target gene.

This is a relevant topic and the methodology is correct to the best of our knowledge. The scope of the study is limited to a specific motif in a bacterial setting, which allows the authors to systematically analyse timing in this system. However, the limited scope isn’t always made clear and generalization could be improved, as suggested below. There are also some points that should be clarified prior to publication.

MAJOR COMMENTS

1) Protein degradation rate. The introduction makes reference to both bacteria and eukaryotic cells (lines 20-30). However, the authors assume a bacterial setting, where the protein degradation rate is determined by cell division. This choice should be made clearly and early in the introduction. To improve generalizability, the authors could include a more thorough examination of how the results depend on the protein degradation rate, since it doesn’t seem like the simulations explicitly incorporate the cell division process. It could also be useful to examine if/how the results depend on the mRNA degradation rate. See also comment 2c.

2) Fig. 2 and associated text.

How is TF occupancy calculated?

How is R defined/calculated? (“TF production rate from a single TF”- line 127)

It would be interesting to compare the results of Fig. 2 to a deterministic model of autoregulation. Would the same conclusions hold for the nonmonotonic relationship between koff and MFPT? Is there any additional insight gained from the stochastic simulations for the MFPT?

Fig. S2 shows the nonmonotonic behaviour as a function of the protein degradation rate. What is the influence of the mRNA degradation rate? Also, it would help to reference panels C and D of this figure near the discussion of Fig.2 in the main text.

3) The analysis presented in Fig. 6 is difficult to follow.

Please clarify what is exactly being done in panels A and B.

Lines 260-261): “In this plot, the number of total TFs is kept constant by adjusting the translation rate to counteract the effect of changing koff. “ Does this mean that in panels A and B koff has the same value for both the autoregulated TF and the target TF, and it changes for both? (if it were only changing for the target, there wouldn´t be need for adjustment. But then weI don´t understand the last paragraph “We further explore the expression timing relationship when the binding sequence for TF and target genes are exactly identical, which means the binding affinity of TFs being identical for both TF and target gene.”) What is the value of alpha for the target gene in panels A, B, C?

Lines 272-273:” The MFPT curve of a target gene regulated by an auto-repressed TF gene is always greater than the MFPT of a target gene under the control of an auto-activated TF gene.” Please specify that this holds only for target repression.

Line 290. “It further provides us with a tool to identify the mode of TF-gene” do the authors mean TF-gene autoregulation?

What happens to the timing noise properties in the situation of Fig. 6? Does it just propagate trivially from the input to the target, or are there parameter combinations that reduce noise for the target compared to the input?

4) The discussion and conclusions are quite short. The implications of the work are made clear, but the relationship to other work in the field on how autoactivation and autorepression change the response time of the system isn’t adequately addressed.

MINOR COMMENTS

1) The use of the word “network” in the Abstract could be misleading, as this creates the expectation that the work will deal with much general gene regulatory networks than the small motif analysed here. We suggest rephrasing and directly using the “single-input module” motif as introduced in line 13.

2) Model: refer to Methods section when introducing the model, as well as in the Figure captions to point to where the rest of the parameter values are found (\\gamma,\\gamma_m_)

3) Notice typo on Fig. S1 (cel-cycle)

4) Fig S3 legend: Is it correct? The filled circles match the power law better than the open circles, whereas open circles are said to be auto-activation.

5) Fig. 5 Please revise the caption.

**Have the authors made all data and (if applicable) computational code underlying the findings in their manuscript fully available?**

Reviewer #1: Yes

Reviewer #2: Yes

Reviewer #3: None

PLOS authors have the option to publish the peer review history of their article (what does this mean?). If published, this will include your full peer review and any attached files.

Reviewer #1: No

Reviewer #2: No

Reviewer #3: No
---

## [Decision Letter · Decision Letter 1]

23 Nov 2021

Dear Professor Brewster,

Thank you very much for submitting your manuscript "Controlling gene expression timing through gene regulatory architecture" for consideration at PLOS Computational Biology. As with all papers reviewed by the journal, your manuscript was reviewed by members of the editorial board and by several independent reviewers. The reviewers appreciated the attention to an important topic. Based on the reviews, we are likely to accept this manuscript for publication, providing that you modify the manuscript according to the review recommendations.

Sincerely,

Alexandre V. Morozov, Ph.D.

Associate Editor

PLOS Computational Biology

Sushmita Roy

Deputy Editor

PLOS Computational Biology

[LINK]

Reviewer's Responses to Questions

**Comments to the Authors:**

Reviewer #1: In the revisions to “Controlling gene expression timing through gene regulatory architecture,” Ali and Brewster focused on broadening the applicability and ultimately the impact of their work, appropriately addressing reviewer questions and adding thoughtful edits. I am happy to recommend acceptance of the manuscript.

The reviewers had challenged the authors with questions about the base assumptions of the model (decay rates), the relevance of the analyses (cross-regulating circuits), and the interpretations of the findings (what does the power law mean?). The authors satisfactorily addressed all points of clarity and adapted their model to answer the questions posed. They found new data to support their reasonings, and in a new set of supplementary figures, defended their work with an investigation of broader parameter sets and modeling schemes. This effort allowed them to draw broader conclusions about the relevance of this work across parameters that may be possessed by a more diverse set of organisms. Notably, these additional analyses and broadening of parameters did alter their conclusions.

While the authors have worked diligently to generate new supporting data to elevate the work, there are still a few lingering points (and minor edits) that the authors can address to further improve the manuscript.

1. While the authors have done a good job in improving the clarity and relevance of their conclusions with some added discussion of intermediate extrema and the interesting phenomena of “cross-regulated” targets, there are points at which the authors stop short of synthesizing the impact of their observations or addressing the limitations that prevent them from drawing such conclusions. Specifically, considering that the discovered power law properties have earned a full main text figure, it would be a lost opportunity not to comment on the impact of this phenomenon. It is understandable that the authors have not been able to pinpoint the underlying reason for this power law, but the mention of the power law without further comment in the added discussion paragraph (Lines 381-390 in the main text) leaves the readers questioning why this power law is mentioned at all.

2. Furthermore, the discussion section could still benefit from more clearly highlighting both the advantages and disadvantages of the presented model, placing it in the context of existing work. Related to the previous comment, if there are limitations of this model that prevent one from drawing precise conclusions, they should be mentioned in the discussion section.

3. Reviewer 1, specific comment #9 was not clearly communicated and seems to have been misunderstood. There is an inconsistency in the text likely stemming from a minor mix-up in the language. Currently, the manuscript reads “The steady state expression level decreases as the binding affinity is increased, from a maximal value of SSE=rb \\/ γ_m γ for a very strong affinity when the promoter is always occupied by a TF to SSE=r_0 b \\/ γ_m γ for very weak activation.” (Lines 156-159 in main text). However, both the plot in Figure 2C and the second half of that sentence conflict with the statement in the first part of that sentence. This should be corrected to either: “The steady state expression level decreases as k_off is increased…” or “The steady state expression level decreases as the binding affinity is decreased…” or some other variation.

4. This was missed in the first round of revisions, but in the caption of Figure 6, the constitutive gene is said to be marked with “black asterisks”, but these appear rendered as filled-in small black circles.

5. Typo: line 321 “To further explore this relationship, In Fig. 6C…” should not have capitalized the word “In”.

Reviewer #2: The authors have addressed the main remarks of mi previous report. There are still a few points that in my opinion can be improved, in particular to improve the clarity of the manuscript and better integrate the outcomes of the model and simulations.

1. Authors: [...] Activating a promoter of a set rate will always make it reach a set level of expression faster than repressing it (in fact, it might not reach that level at all if you add repression). Using a relative threshold is a way to normalize two different scenarios to measure the dynamics of a promoter that reaches a set level of expression. As an example, the famous result that autorepression “speeds up” the response of a gene is only true if you take care to normalize the promoter such that the levels are the same (or to use a threshold).

I do understand that this is what it has been done in the literature, I was questioning about the biological relevance (since it should be the absolute TF concentration that matters, not the one relative to ss). This was however more a comment than a question, I do not know if the authors wish to discuss that, but I believe that it would help.

2. Authors: [...] As for the origin of the power-law dependencies, we have spent considerable time on this and still don’t have a satisfactory explanation. We have tried to answer through toy models or exact solutions but have not had much success. [...]

I am a bit confused, and took me some time trying to understand. What about the new supplementary information S2 and Fig.S11? I know that this is not an exact solution, but it is still probably possible to improve our understanding from that. The SI should probably be valued more in the main text, and discussed. See also point below.

3. Authors: [...] The model can be solved exactly to get an analytical

expression for response time (time to reach a certain threshold, analogous to MFPT). The

analytical solution recapitulates all the features for an autotregulatory gene. However, we

could not get a simplified analytical form for peak response time, even for this simplified

model.

Again, reading the caption of S11: "(B) Peak MFPT versus alpha as well as koff at peak versus alpha

(inset) from the toy-model matches well with the full deterministic model." This is not an analytical result but it shows that the toy model developed captures the power-law-like behaviour at least for the parameters chosen. I imagine it should then be possible to show that, with some approximations, by setting the derivative with respect k_off = 0 one obtains alpha^x. Otherwise, how the simulations compare to the toy model?

It is (at least for me) very confusing (and time consuming) trying to understand the discussion on page 6, and the addition of $R = b r_0 + b(...$ is rather misleading. This approximation cannot infact hold as a function of time (it is assumed that n_TF = 1 at all times) and if I am not wrong, when computing the passage time to get to half of the SSE for a deterministic model dP/dt = R-\\gamma P one would get the response time for a constitutive gene, \\tau_cons in the SI.

Instead of this part, I would suggest the authors to include here the results of the calculations done in the SI, and if/when possible, to better compare the results shown in Fig.3 to the model of the current SI.

**Have the authors made all data and (if applicable) computational code underlying the findings in their manuscript fully available?**

Reviewer #1: Yes

Reviewer #2: Yes

PLOS authors have the option to publish the peer review history of their article (what does this mean?). If published, this will include your full peer review and any attached files.

Reviewer #1: No

Reviewer #2: No

Figure Files:

Data Requirements:

Reproducibility:

References:

---

## [Editor Report · Decision Letter 2]

8 Dec 2021

Dear Professor Brewster,

We are pleased to inform you that your manuscript 'Controlling gene expression timing through gene regulatory architecture' has been provisionally accepted for publication in PLOS Computational Biology.

Best regards,

Alexandre V. Morozov, Ph.D.

Associate Editor

PLOS Computational Biology

Sushmita Roy

Deputy Editor

PLOS Computational Biology

---

## [Editor Report · Acceptance letter]

27 Dec 2021

PCOMPBIOL-D-21-00855R2 

Controlling gene expression timing through gene regulatory architecture

Dear Dr Brewster,

I am pleased to inform you that your manuscript has been formally accepted for publication in PLOS Computational Biology. Your manuscript is now with our production department and you will be notified of the publication date in due course.

With kind regards,

Anita Estes
